# Evo-PI: Scaling Medical Reasoning via Evolving Principle-Guided Reinforcement Learning

## Abstract

Effective reasoning over complex visual data and medical knowledge is critical for medical Visual Question Answering (VQA). While multimodal large language models (MLLMs) show promise, their reasoning capabilities remain fundamentally capped by the static nature of current training paradigms. Existing reinforcement learning (RL) methods act as fixed tutors, providing unchanging guidance that often optimizes output format without explicit medical expertise, leading to performance plateaus and reward hacking. Drawing inspiration from how human experts continuously refine clinical principles, we introduce **Evo-PI**, a framework that operationalizes a synergistic loop of evolving principle-guided learning. Evo-PI generates, applies, and iteratively refines abstract medical principles, which serve as dynamic rewards. This co-evolution of the reasoning model and its guiding principles enables MLLMs to develop more robust and clinically aligned reasoning. Across eight medical VQA benchmarks, Evo-PI consistently improves performance over diverse backbones and RL algorithms, achieving up to 24.6% accuracy gains. Our results establish evolving principle scaling as a scalable and generalizable paradigm for aligning MLLMs with expert-like reasoning, advancing the path toward trustworthy medical AI.

## 1 Introduction

Multi-modal Large Language Models (MLLMs) have demonstrated significant potential for medical applications, particularly in Visual Question Answering (VQA) tasks Yin et al. (2023); Xiao et al. (2025). For instance, MLLMs can analyze X-ray images to aid clinicians in the rapid screening of potential cases, as seen during the COVID-19 pandemic. Such models not only assist with preliminary diagnostic tasks but also help analyze and infer underlying pathologies, enabling clinical workflows to fully leverage the knowledge embedded within MLLMs Ye & Tang (2025).

The capacity of existing MLLMs to perform specialized medical tasks is driven by their sophisticated reasoning capabilities. Echoing the development of general-purpose models Comanici et al. (2025); Team (2023); Bai et al. (2025), these reasoning abilities are primarily honed through large-scale pre-training and subsequent fine-tuning on domain-specific data Chen et al. (2024); Wu et al. (2025); Pan et al. (2025); Lai et al. (2025).

A predominant strategy for enhancing the reasoning capabilities of medical MLLMs is to expand the training dataset with more medical VQA samples, following established scaling laws Chen et al. (2024). However, this approach faces a significant bottleneck, as acquiring accurately annotated medical data requires labor-intensive review by clinical experts. Consequently, the cost of curating high-quality training data for medical MLLMs is exceptionally high compared to other specialized domains Wang et al. (2024); Liu et al. (2024).

Recent methods using reinforcement learning (RL) have proved effective in enhancing the reasoning capabilities of MLLMs beyond what is achieved with Supervised Fine-Tuning (SFT) Lai et al. (2025); Chen et al. (2024). Compared to SFT, RL-based methods can improve the generalization performance of medical MLLMs Lai et al. (2025); Pan et al. (2025); Wu et al. (2025). Many of these techniques are inspired by Group Relative Policy Optimization (GRPO) Shao et al. (2024), which employs both accuracy and format compliance as reward signals to constrain the model's output to a structured response, such as a chain of thought followed by a final answer Guo et al. (2025).

Despite this progress, existing medical MLLMs still falter on complex clinical reasoning. We argue this stems from a fundamental limitation in current RL paradigms: they focus on refining reasoning format without explicitly enhancing the model's underlying medical knowledge. A paradigm that merely constrains reasoning paths to fit a desired structure is unlikely to replicate the rapid capability scaling observed in general-purpose models Lockyer et al. (2017). While advanced RL techniques (e.g., clip-higher, GSPO Yu et al. (2025); Chen et al. (2025); Zheng et al. (2025)) can yield marginal improvements, these optimizations alone are insufficient for models to approach the upper bounds of their reasoning potential in medicine.

This paradigm starkly contrasts with how human experts learn. Medical students first acquire basic clinical principles, then continuously refine them by generalizing from diverse case studies, enabling transfer to complex and unfamiliar situations Issa et al. (2011). Inspired by this dynamic and scalable learning process, we propose a novel framework that operationalizes this principle of co-evolution.

This evolving guidance mechanism offers two key advantages: (1) it applies the principles of scaling laws directly to the reward model, providing theoretical grounding for its efficacy Kaplan et al. (2020); and (2) it establishes a synergistic process, where evolving principles deepen medical knowledge integration while simultaneously mitigating reward hacking Skalse et al. (2022). Figure 1 illustrates the key insights of our idea.

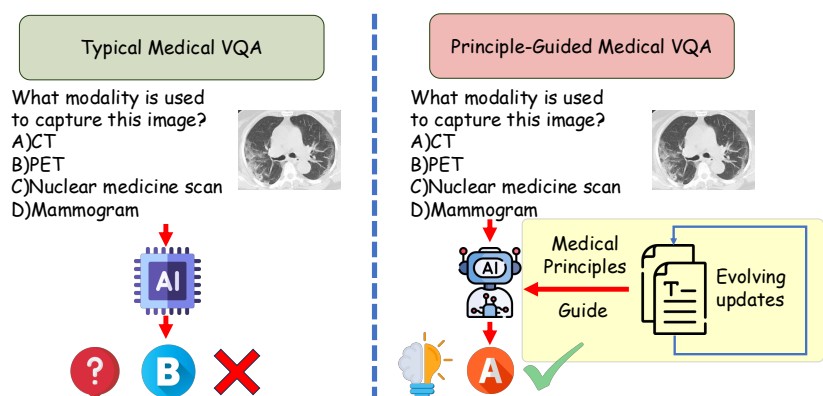

Figure 1: Comparison of typical Medical VQA with our Principle-Guided framework. (Left) Without explicit guidance, a standard MLLM makes a superficial guess based on visual similarity, resulting in a reasoning failure. (Right) Our framework equips the MLLM with evolving medical principles as a clinical guide, fostering a robust reasoning process that correctly identifies the modality, systematically checks for abnormalities, and arrives at the correct answer through a transparent and clinically aligned pathway.

To this end, we introduce **Evo-PI**, an **Evol**ving **P**rinciple-guided reinforcement learning framework for medical reasoning. Evo-PI operationalizes this synergistic loop through three phases: (1) *Principle bank initialization*, where a knowledgeable LLM distills an initial set of principles from case examples; (2) *Principle-Guided RL*, where a backbone MLLM is trained with dynamic rewards derived from these principles, as evaluated by a frozen judge LLM; and (3) *Principle Evolution*, where the knowledgeable LLM refines and expands the principle set based on training dynamics, preparing a more sophisticated guide for the next iteration.

In summary, our contributions are:

- We propose Evo-PI, the first framework to our knowledge that enables co-evolution between a reasoning model and its guiding medical principles, moving beyond the fixed paradigms of existing RL methods.
- Evo-PI mimics principles learning in medical training, enhancing and stimulating the intrinsic knowledge of medical MLLMs.
- We introduce evolving principle scaling, where abstract medical knowledge is progressively refined and expanded to serve as a dynamic and scalable reward signal, effectively mitigating reward hacking.

- Through extensive experiments across eight medical VQA benchmarks, we show that Evo-PI delivers consistent and significant performance gains across diverse MLLM backbones.

## 2 RELATED WORK

**General MLLMs and Medical MLLMs**   Recent years have seen the emergence of large-scale MLLMs, such as GPT-4o Team (2023) and the Gemini series Comanici et al. (2025). These models acquire strong cross-modal reasoning by leveraging vast data collections, but consistently underperform on tasks requiring deep domain expertise. Medical MLLMs address this gap, typically through SFT on expert-annotated datasets like PubMedVision Chen et al. (2024). While effective, such approaches incur substantial annotation costs, creating a major scaling bottleneck. To mitigate data scarcity, recent works have explored RL for post-training alignment Pan et al. (2025); Lai et al. (2025); Wu et al. (2025). However, these methods generally treat medical knowledge as static resources for fine-tuning rather than as dynamic, guiding components of the learning process. As a result, they rely on fixed reward heuristics and largely overlook explicit medical knowledge, even though knowledge-aware judging has proven effective in general LLM settings Zheng et al. (2023). This gap highlights a critical need to rethink how medical expertise is integrated into the alignment process.

**Reinforcement Learning Algorithms for Post-Training**   State-of-the-art alignment commonly employs RL algorithms such as PPO Schulman et al. (2017) and GRPO Shao et al. (2024), with subsequent variants like GSPO Zheng et al. (2025) addressing challenges including training instability and entropy collapse Cui et al. (2025). A common limitation across these techniques is their reliance on reward functions with fixed, pre-defined strategies. Such rigidity leaves models susceptible to reward hacking Skalse et al. (2022), particularly in knowledge-intensive domains like medicine, where they may exploit surface-level heuristics of the reward scheme rather than acquiring deeper, clinically aligned reasoning skills. These limitations motivate the central hypothesis of our work: to foster robust medical reasoning, the reward mechanism itself must be *dynamic, knowledge-driven, and co-evolve with the model being trained*.

## 3 EVO-PI

Current training paradigms lack a mechanism to transform medical principles into dynamic guidance and reward signals that evolve with the model. To address this gap, we introduce **Evo-PI**, an iterative framework that *generates, applies, and refines* domain principles to strengthen the reasoning capabilities of MLLMs on complex medical VQA tasks.

### 3.1 PROBLEM DEFINITION

In medical VQA, a MLLM answers text questions about a medical image. Formally, given a medical MLLM $\mathcal{M}_\theta$ with its parameters $\theta$, the input is a set of medical VQA questions. Each medical VQA question $q$ consists of a medical image $I$ and a textual question $T$, which can be represented as:

$$q = (I, T), \quad I \in \mathbb{R}^{H \times W \times C}, \ T = (w_1, w_2, \ldots, w_n), \ w_i \in \mathcal{V}, \tag{1}$$

where the textual question $T$ is represented as a sequence of $n$ tokens, with each token $w_i$ drawn from the vocabulary $\mathcal{V}$. $H$, $W$, and $C$ represent the height, width, and channel dimension of the image, respectively.

The model predicts a textual answer $\hat{a}$ to the medical question, represented as a sequence of tokens: $\hat{a} = (y_1, y_2, \ldots, y_m), y_j \in \mathcal{V}$, where the answer $\hat{a}$ consists of $m$ tokens, and each token $y_j$ is drawn from the same vocabulary $\mathcal{V}$. The training objective is to maximize the conditional likelihood of the answer given the input, defined as follows:

$$\max_\theta \ \log \mathcal{M}_\theta(\hat{a} \mid I, t) = \max_\theta \sum_{j=1}^{m} \log \mathcal{M}_\theta(y_j \mid y_{<j}, I, t). \tag{2}$$

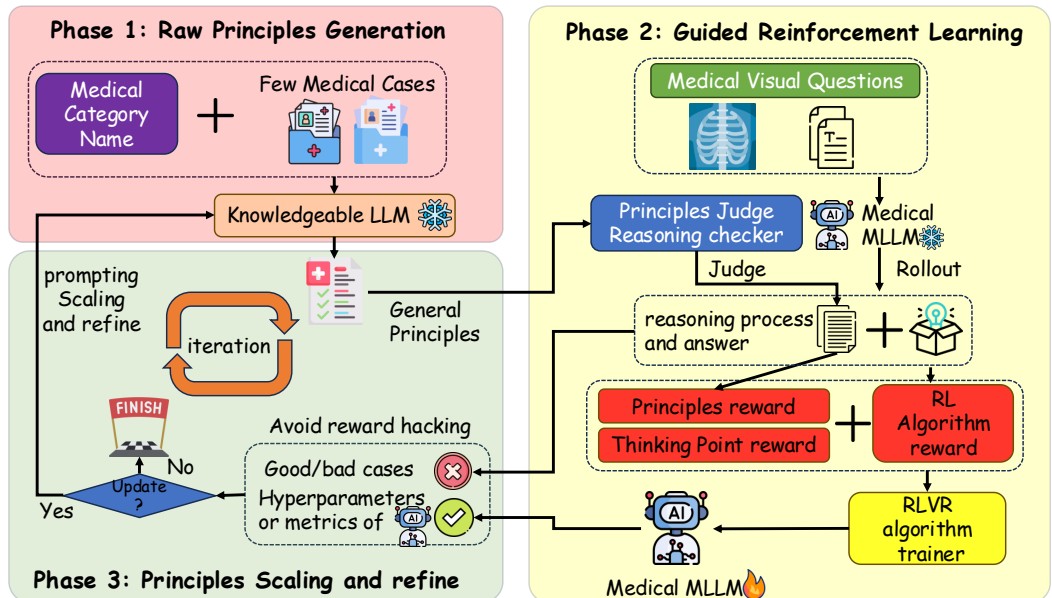

Figure 2: Overview of the **Evo**ving **P**rinciple-guided **I**terative framework (Evo-PI).

## 3.2 OVERALL FRAMEWORK

Our proposed framework, Evo-PI, directly addresses the gaps in existing work by externalizing medical knowledge into editable principles, optimizing these principles as learning time signals rather than a fixed knowledge set, and continuously revising them to balance exploration and exploitation during principle scaling and refinement. As illustrated in Figure 2, Med PI runs a three-stage loop. (1) **Principle bank initialization**: We begin by creating a set of fundamental principles for the medical VQA task. This is achieved by prompting a frozen knowledgeable LLM with a few-shot medical cases and category names to distill an initial set of general principles that capture clinically grounded heuristics for image and text reasoning. (2) **Guided Reinforcement Learning**: A backbone MLLM is trained with RL to answer medical visual questions. For each instance, the model generates an answer and a corresponding reasoning chain in a process known as a *rollout*. A separate, frozen judge LLM evaluates adherence to the relevant principles, yielding a *principle reward* and a *thinking point reward*. These are combined with the *base RL environment reward* to update the parameters of the MLLM backbone. (3) **Principle Evolution**: At the end of each iteration, if the stopping criteria are not met, the Knowledgeable LLM from the first stage is prompted to scale and refine the set of principles, guided by the current MLLM's performance. The next iteration then proceeds with the updated principles. The process terminates once a stopping condition is satisfied.

## 3.3 PRINCIPLE BANK INITIALIZATION

Unlike prior work that scales data or fixes heuristic rewards without updating knowledge, Evo-PI externalizes clinical knowledge and initializes an editable principle bank. Inspired by clinical training, where protocols are distilled and revised through repeated case review, this stage converts tacit medical knowledge into editable rules that supervise learning. Specifically, given a medical VQA dataset with type annotations $\mathcal{C}$, we sample a small anchor set of question-answer pairs $\mathcal{Q}_C$ for each medical type $c \in \mathcal{C}$. Next, Evo-PI deployed a frozen Knowledgeable LLM($\mathcal{M}_K$) first to generate a set of candidate principles $P_c$ conditioned on a medical type name $c \in \mathcal{C}$ (e.g., MR, CT), for all $c \in \mathcal{C}$. Subsequently, $\mathcal{M}_K$ is then prompted with a small random set of question–answer pairs $\mathcal{Q}_C$ drawn for type c as context, and refines the principles $P$ for coverage and specificity. The initial principle bank is denoted as $P = \bigcup_{c \in \mathcal{C}} P_c$. The resulting principles are stored for downstream use. Prompt templates for both initial generation and iterative evolution are provided in Appendix A.5.

We use a text LLM rather than an MLLM at this stage because the goal is medical plausibility, rather than visual grounding. This decouples principle creation from the backbone and modality, requires no shared architecture or pre-training corpus, and avoids alignment constraints.

### 3.4 GUIDED REINFORCEMENT LEARNING

The objective of this stage is to train the medical MLLM to answer visual questions by converting editable principles into reward signals, overcoming fixed heuristic rewards and the lack of knowledge updates in prior work. For each medical VQA instance, the backbone MLLM performs a rollout to produce a reasoning trace and an answer. A frozen judge LLM ($\mathcal{M}_J$) evaluates adherence to the current principle set and returns principle based rewards. These are merged with the base RL reward, and the composite signal is used to update the parameters of this medical MLLM ($\mathcal{M}_\theta$).

Specifically, given a medical VQA input $q = (I, T)$, Evo-PI prompts the learnable medical MLLM $\mathcal{M}_\theta$ with an answer format that first elicits a step-wise reasoning trace $rt$ and then a final answer $\hat{a}$ based on this reasoning trace $rt$. The medical MLLM ($\mathcal{M}_\theta$) completes one rollout $o = (rt, \hat{a})$ or more $\{o_i\}_{i=1}^G$, depending on the chosen RL algorithm. The LLM judge ($\mathcal{M}_J$) scores $o$ against the principles to produce a reward for the principles and a reward for the point of thought, which is combined with the environment reward to update $\theta$. The detailed prompt templates can be found in the Appendix A.7.

**Principles Judge and Principle Rewards**    Med PI concatenates the principles $P$ from the previous stage into a description paragraph $D = \bigoplus_{i=1}^n P_i$, where $\oplus$ denotes the concatenation operation. A frozen principles judge LLM ($\mathcal{M}_J$) compares the description paragraph $D$ with the reasoning trace $rt$, and returns a count $c$ of satisfied principles. The principle reward can be formally defined as:

$$\text{Reward}_P = \begin{cases} \dfrac{c}{|P|}, & 0 \le c \le |P|, \\ 0, & c < 0 \text{ or } c > |P|, \end{cases} \tag{3}$$

where the constraint $c > 0$ is designed to prevent MLLM from engaging in reward hacking by releasing large amounts of repetitive content in the reasoning traces $rt$. In contrast, $c < 0$ reflects the error of the judge from $\mathcal{M}_J$ and is neutralized by clipping. The principle rewards are normalized by $P$ and clip out-of-range values. Simultaneously, the same frozen judge ($\mathcal{M}_J$) verifies, step by step, that the reasoning trace $rt$ supports the final answer $\hat{a}$, analogous to a clinician-style verification workflow.

**Thinking Point Reward**    In this step, Evo-PI extracts the reasoning trace enclosed by the `<think> </think>` tags from each model response and checks whether the trace satisfies the enumerated points of reasoning. A reasoning checker parses the trace, identifies bullet indicators using regular expressions, and counts the number bullet points $b$ that are correctly addressed. The thinking point reward is then $\text{Reward}_T$ defined as follows:

$$\text{Reward}_T = \begin{cases} \dfrac{b}{|P|}, & 0 \le b \le |P|, \\ 0, & b > |P|, \end{cases} \tag{4}$$

where $\text{Reward}_T$ is normalized by the number of listed points and the condition $b > 0$ is also used to prevent reward hacking.

**RLVR Training Procedure**    After obtaining reward signals from the Principles Reward ($\text{Reward}_P$) and Thinking Point Reward ($\text{Reward}_T$), Evo-PI treats the backbone MLLM as a policy and optimizes it with Reinforcement Learning with Verifiable Rewards (RLVR) Wen et al. (2025). Specifically, Evo-PI adopts GRPO Shao et al. (2024) and GSPO Zheng et al. (2025) as the RLVR trainer. For example, In GRPO, the per-sample scalar return $r_i$, is formed from the verifiable rewards and the environment signal. The normalized per-token advantage is defined as follows:

$$\hat{A}_{i,t} = \frac{r_i - \mu(\{\text{Reward}_P, \text{Reward}_T, \text{Reward}_{RLVR}\})}{\sigma(\text{Reward}_P, \text{Reward}_T, \text{Reward}_{RLVR})}. \tag{5}$$

where $\text{Reward}_{RLVR}$ is the original reward defined by the RLVR algorithm (e.g., format reward and accuracy reward in GPRO). Accordingly, the policy ratio is defined as follows:

$$r_{i,t}(\theta) = \frac{\mathcal{M}_\theta\big(o_{i,t} \mid q, o_{i,<t}\big)}{\mathcal{M}_{\theta_{\text{old}}}\big(o_{i,t} \mid q, o_{i,<t}\big)}. \tag{6}$$

where $o_i \in$ denotes the $i$-th rollout (i.e., reasoning trace $rt_i$ and answer sequence $\hat{a}_i$) generated by the medical MLLM $\mathcal{M}_\theta$ for query $q$, with $G$ total rollouts sampled per query. The final GRPO objective used to update the $\theta$ combines the principle and thinking point signals with the environment reward to promote stable policy improvement, and is given by:

$$\mathcal{J}_{\text{Evo-PI}}(\theta) = \mathbb{E}_{\substack{(q,A)\sim\mathcal{D} \\ \{o_i\}_{i=1}^{G}\sim\mathcal{M}_{\theta_{\text{old}}}(\cdot|q)}} \left[ \frac{1}{G} \sum_{i=1}^{G} \frac{1}{|o_i|} \sum_{t=1}^{|o_i|} \min\Big( r_{i,t}(\theta)\hat{A}_{i,t}, \text{clip}\big(r_{i,t}(\theta), 1-\varepsilon, 1+\varepsilon\big)\hat{A}_{i,t} \Big) \right]. \tag{7}$$

## 3.5 PRINCIPLE EVOLUTION

Phase three consists of three core components to adapt the principle set by controlling the iteration, scaling the principles, and refining them, rather than relying on fixed rewards or static knowledge during training.

**Iteration and Generalization Control** To prevent overfitting to cases produced during training and limit drift from medical knowledge, we cap the number of loop passes and apply early stopping on a held-out validation set.

**Balance of Exploitation and Exploration** We track token-level policy entropy during the process of reinforcement learning. Training terminates typically under two conditions: *Entropy Collapse* Cui et al. (2025) and *Abnormal Entropy Increase*. Entropy collapse is a common phenomenon indicating a reduction in the MLLM's exploration space, as the model tends to become more certain of its own inferences, also known as exploitation over exploration. High entropy indicates that the policy prioritizes exploration to discover solutions, whereas low entropy signifies a focus on consistently exploiting optimal actions. In Evo-PI, medical MLLMs serve as policy models that are updated by the RLVR trainer, where the balance of exploring diverse case patterns with reliably solving individual cases makes entropy a suitable stopping signal. Because principles are embedded in the judge and never exposed to the MLLM, reward hacking is curtailed, though mild entropy rises can occur. In practice, Evo-PI terminates training if entropy falls to a collapse threshold or exceeds the baseline recorded after the first MLLM update.

**Scaling and Refinement of Principles** If termination criteria are not met, a frozen Knowledgeable LLM receives the current principles from the latest round and prompts it to scale and refine these principles. The update increases coverage while keeping rules abstract, concise, and compositional, where redundant rules are merged, low utility rules are pruned, and new numbered rules are added only for recurring failure modes. Prompts for this step are given in Appendix A.6.

## 4 EXPERIMENTS

We conduct quantitative and qualitative experiments to evaluate Evo-PI.

### 4.1 EXPERIMENTAL SETTING

**Datasets** We conduct experiments on the OmniMedVQA dataset Hu et al. (2024), which is used in Med-R1 and designed for the medical VQA task. OmniMedVQA spans eight imaging modalities: Computed tomography (CT), dermoscopy (DER), fundus photography (FP), microscopy images (MI), magnetic resonance imaging (MR), optical coherence tomography (OCT), ultrasound (US), and X-ray. After removing duplicate cases, we split the data into training, testing, and validation sets in an 8:1:1 ratio. Table 1 summarizes the statistics of these eight benchmark datasets.

**Hyperparameter Settings** Evo-PI employs a consistent set of hyperparameters across all eight datasets. The default iteration number is three, with one training epoch in each iteration. The batch

Table 1: Statistics of the datasets.

| Dataset | CT | DER | FP | MI | MR | OCT | US | X-ray |
|---|---|---|---|---|---|---|---|---|
| Training | 12,647 | 5,343 | 4,318 | 4,544 | 25,501 | 3,716 | 8,792 | 6,332 |
| Validation | 1,581 | 668 | 540 | 568 | 3,188 | 465 | 1,099 | 792 |
| Test | 1,581 | 668 | 540 | 568 | 3,188 | 465 | 1,100 | 792 |
| Total | 15,809 | 6,679 | 5,398 | 5,680 | 31,877 | 4,646 | 10,991 | 7,916 |

size is 256 for all backbone medical MLLMs. For group-based RL algorithms (such as GRPO), the rollout number defaults to 8. For RL algorithms using cilp-higher (e.g., GSPO), Evo-PI set the `clip_ratio_low`=0.0003 and `clip_ratio_high`=0.0004.

We recommend knowledgeable and judge LLMs with at least 7B parameters. In our setup, GPT-4o-mini[1] serves as the knowledgeable LLM for principle generation, scaling, and refinement, and Qwen2.5-7B-Instruct[2] serves as the judge LLM to evaluate the responses from medical MLLM. We implemented parallelism using FSDP on the Verl training framework. All backbone models are trained on 4×H100 80GB SXM GPUs, and the judge LLM runs on a separate H100 80GB GPU.

**Baselines and Evaluation Metrics**   We evaluate Evo-PI against two strong medical VQA backbones, HuatuoGPT-Vision  Chen et al. (2024) and Med-R1 Lai et al. (2025), both state-of-the-art on medical VQA. For each backbone, we report Accuracy before and after applying Evo-PI under identical decoding and data splits. We also conduct a horizontal comparison across base MLLM variants of varying sizes corresponding to the backbones' underlying models.

Table 2: Overall comparison on the medical visual question answering task. **Bold** indicates the best results. HuatuoGPT-Vision is based on Qwen 2.5-VL-7B. Med-R1 comprises eight modality-specific submodels for the eight medical modalities, each based on Qwen 2-VL-2B.

| Dataset | CT | DER | FP | MI | MR | OCT | US | X-ray | Average |
|---|---|---|---|---|---|---|---|---|---|
| **Qwen 2-VL-2B [1]** | 0.4023 | 0.4177 | 0.4685 | 0.4208 | 0.4410 | 0.3828 | 0.3864 | 0.4823 | 0.4252 |
| **Qwen 2-VL-7B [2]** | 0.6818 | 0.5719 | 0.7444 | 0.6250 | 0.5452 | 0.6323 | 0.3827 | 0.7197 | 0.6129 |
| **Qwen 2-VL-72B [3]** | 0.6797 | 0.6531 | 0.7258 | 0.6784 | 0.6939 | 0.7276 | 0.5139 | 0.7211 | 0.6805 |
| **Qwen 2.5-VL-3B [4]** | 0.7103 | 0.6078 | 0.6981 | 0.5996 | 0.5364 | 0.6839 | 0.4409 | 0.7462 | 0.6278 |
| **Qwen 2.5-VL-7B [5]** | 0.6736 | 0.7006 | 0.7056 | 0.6004 | 0.5533 | 0.5570 | 0.3355 | 0.7563 | 0.6103 |
| **Qwen 2.5-VL-72B [6]** | 0.6618 | 0.6975 | 0.7104 | 0.6937 | 0.6364 | 0.6922 | 0.6985 | 0.7981 | 0.6771 |
| **LLaVA-Med [7]** | 0.1869 | 0.4495 | 0.3903 | 0.3329 | 0.2747 | 0.3461 | 0.2988 | 0.3068 | 0.3233 |
| **RadFM [8]** | 0.2756 | 0.3921 | 0.3686 | 0.2797 | 0.2406 | 0.3280 | 0.1657 | 0.3095 | 0.2950 |
| **Med-Flamingo [9]** | 0.3128 | 0.4856 | 0.4126 | 0.3003 | 0.2634 | 0.2516 | 0.3169 | 0.4401 | 0.3429 |
| **MedVInT [10]** | 0.4074 | 0.2911 | 0.3184 | 0.3202 | 0.4310 | 0.2326 | 0.4126 | 0.5510 | 0.3705 |
| **HuatuoGPT-Vision [11] (Base [5])** | 0.6534 | 0.6841 | 0.7630 | 0.7130 | 0.6866 | 0.7763 | 0.4818 | 0.8005 | 0.6948 |
| **Evo-PI (Base [11] + GSPO)** | 0.8797 | 0.9021 | **0.9369** | **0.9580** | 0.9138 | 0.9824 | 0.9109 | **0.9367** | 0.9295 |
| **Evo-PI (Base [11] + GRPO)** | 0.8797 | 0.9021 | **0.9369** | **0.9580** | 0.9418 | 0.9824 | 0.9109 | **0.9367** | 0.9295 |
| **Relative Gains (%)** | 22.63%↑ | 23.70%↑ | 18.91%↑ | 24.50%↑ | 25.52%↑ | 20.61%↑ | 47.28%↑ | 13.62%↑ | 24.59%↑ |
| **Med-R1 [12] (Base [1])** | 0.7160 | 0.8338 | 0.9019 | 0.7447 | 0.5144 | 0.8946 | 0.7773 | 0.7854 | 0.7710 |
| **Evo-PI (Base [12] + GSPO)** | **0.9628** | **0.9334** | 0.9081 | 0.8739 | 0.9298 | **0.9934** | **0.9982** | 0.9038 | 0.9391 |
| **Evo-PI (Base [12] + GRPO)** | 0.9676 | 0.9319 | 0.9195 | 0.8720 | **0.9462** | 0.9890 | **0.9982** | 0.9101 | **0.9412** |
| **Relative Gains (%)** | 25.16%↑ | 9.96%↑ | 1.76%↑ | 12.92%↑ | 43.18%↑ | 9.88%↑ | 22.09%↑ | 12.47%↑ | 17.18%↑ |
| **Relative Gains (Task)** | 23.89%↑ | 16.83%↑ | 10.34%↑ | 18.71%↑ | 34.35%↑ | 15.24%↑ | 34.68%↑ | 13.05%↑ | 20.88%↑ |

## 4.2   OVERALL PERFORMANCE COMPARISON

The main quantitative results for the Medical VQA task are presented in Table 3, from which several key observations can be drawn. First, our proposed framework, **Evo-PI**, consistently and significantly improves the backbone MLLMs across datasets and RL algorithms. Specifically, Evo-PI yields improvements of 23.89%, 16.83%, 10.34% 18.71%, 34.35%, 15.24%, 34.68%, and 13.05% in accuracy over the respective tasks, demonstrating its robustness and effectiveness in medical VQA

---

[1] https://platform.openai.com/docs/models/gpt-4o-mini
[2] https://huggingface.co/Qwen/Qwen2.5-7B-Instruct

task. On the HuatuoGPT-Vision model at level 7B, using Evo-PI can improve the accuracy of the backbone model by an average of 24.59%. In particular, even in the Med-r1 serious models trained using GRPO, our Evo-PI achieves an average 17.18% boosted using the same GRPO trainer.

Finally, on OCT and ultrasound, the Evo PI enhanced backbones exceeded 99.3% for the first time, indicating that principled guidance can push task performance toward clinical utility.

### 4.3 IMPACT OF ITERATION DESIGN.

In Evo-PI, the guiding principles are updated at each iteration. We investigate the respective effects of each factor under an iterative mechanism.

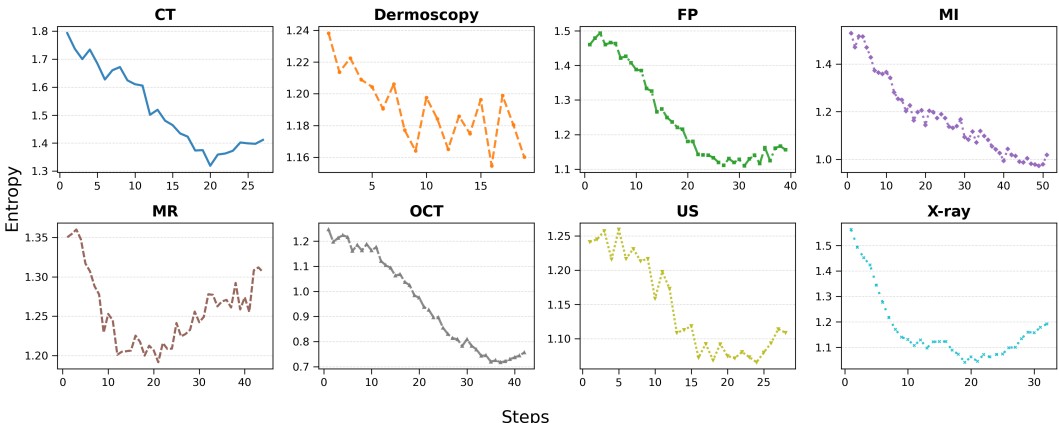

Figure 3: Entropy dynamics for eight independent training runs, each with its own step axis. Each subplot corresponds to a modality: CT, DER, FP, MI, MR, OCT, US, and X-ray. All backbones are from the Med-R1 series, and all runs use GRPO.

**Principles Cases in Iteration**  At each iteration, the knowledgeable LLM initializes or updates the current set of principles, enabling direct observation of how they evolve.

We observe two general phenomena. (i) When the initial set is incomplete, the knowledgeable LLM refines or decomposes existing principles while introducing additional ones that better meet the task requirements. (ii) When the initial set is already reasonably comprehensive, the knowledgeable LLM performs fine-grained edits, improving individual principles one by one.

Taking the principles for X-ray as an example, we demonstrate how they are generated and how they are iteratively scaled and refined in Appendices A.5 and A.6, respectively.

**Entropy Analysis in Iteration**  Entropy quantifies the exploration–exploitation balance of a policy, offering an interpretable lens on learning dynamics Sutton (1988); Cui et al. (2025). High entropy typically signals exploratory behavior to discover new solutions, whereas low entropy indicates confident, exploitative behavior with more consistent action selection.

In Evo-PI, medical MLLMs are optimized as policies under an RLVR trainer. They must balance exploring diverse case patterns with reliably solving individual cases. This motivates using entropy-based stopping: we terminate an iteration when entropy evidences either collapse (i.e., over exploitation) or abnormal increase (i.e., unstable exploration), marking ineffective learning progress. As shown in Figure 3, entropy decreases across all modalities, with low amplitude fluctuations near the end of training that reflect the evolving exploration–exploitation balance under Evo-PI.

**Ablation on Principles Gains and Iteration Gains**  The gains derived from principles and iteration demonstrate how the principles within iterative process exert their influence in Evo-PI. We conducted ablation experiments on the results for Med-R1 when using GRPO as the RLVR algorithm.

We find that applying the principles yields an average improvement of 13.48%. On top of this, introducing iteration provides an additional performance boost of around 3.7%. Detailed ablations are provided in Appendix A.9.

## 4.4 QUALITATIVE STUDIES

We conducted tests using Evo-PI across all modalities and observed marked improvements in reasoning quality: post-training traces become more coherent, clinically grounded, and logically ordered. We attribute the performance gains primarily to this enhancement in stepwise reasoning rather than superficial answer heuristics. A representative CT case is shown below:

---

**Case Study: CT Scan**

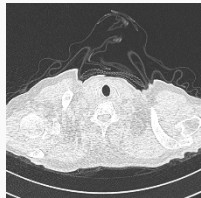

**Problem:** Is there any perceptible anomaly present in this picture?
   A)No    B)Yes
**Ground Truth:** A

**Med-R1's answer**:
<think> The image description A grayscale image of a cross-sectional view of a human head, showing the brain, skull, and surrounding tissues. The image appears to be a medical scan, possibly an MRI or CT scan, with various shades of gray representing different tissue densities. </think> <answer>B</answer>
**Evo-PI's answer (evolved)**:
<think> 1. The image provided is a CT scan of the neck, which is typically used to visualize soft tissues and structures within the neck.
2. The scan shows a cross-sectional view of the neck, including the thyroid gland, which is a critical structure in the neck.
3. The thyroid gland appears to be in its normal position, with no visible signs of enlargement or abnormal shape.
4. There are no visible masses, calcifications, or other abnormalities that would suggest a pathological condition.
5. The surrounding structures, such as the trachea and carotid arteries, appear to be in their normal anatomical positions.
6. The image does not show any signs of inflammation, infection, or other conditions that would typically be indicated by abnormal findings on a CT scan.
Based on the above observations, there does not appear to be any perceptible anomaly present in the image.
</think> <answer>A</answer>

---

From this example, the Evo-PI enhanced answer exhibits stronger logical coherence and makes fuller use of CT-specific knowledge than the original Med R1 output, leading to the correct solution. Additional case studies are provided in Appendix A.8.

## 5 CONCLUSION

We propose Evo-PI, a framework that enhances reasoning in medical VQA by iteratively generating and refining guiding principles. Our approach uses a judge model to convert these principles into a dynamic reward signal, which integrates deep medical knowledge while mitigating reward hacking. Experiments show Evo-PI robustly boosts performance and produces more transparent reasoning, advancing the development of trustworthy medical AI.

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

# A APPENDIX

## A.1 THE GUIDELINE OF APPENDIX

The appendix is organized as follows:

- In section A.2, we introduce Ethics Statement of our paper.
- In section A.3, we introduce Reprodicibility Statement of our paper.
- In section A.4, we introduce how we use the LLM in our work as required.
- In section A.5, we introduce how we generate and scaling principles for each sub-modality of medical questions.
- In section A.6, we introduce how these principles are scaling and refined.
- In section A.7, we introduce the post-training prompt to control the rollout process.
- In section A.8, we introduce the case study across different datasets.
- In section A.9, we further introduce some ablation study about the impact of the iterations and the principles.

## A.2 ETHICS STATEMENT

All datasets used in this research are publicly available and were sourced from previous studies that have undergone appropriate ethical review. Our work did not involve the collection of any new data from human subjects. We have adhered to all data usage agreements and licenses associated with these pre-existing datasets.

### A.3 Reprodicibility Statement

We are committed to making our research reproducible. All datasets used in this study are publicly available, and we provide detailed descriptions and sources in our experimental setup section. The source code for our proposed framework, Evo-PI, including all scripts required to reproduce the experiments and analyses presented in this paper, is available at an anonymized code repository: https://anonymous.4open.science/r/Evo-PI-ECB4.

### A.4 Usage of LLM

This paper primarily employs large language models (LLMs) to refine the overall quality of writing, with a particular focus on eliminating incorrect expressions, minimizing grammatical errors, and enhancing clarity, coherence, and readability to ensure the text meets standards of ICLR.

### A.5 The prompt of generate and scaling raw principles

In this section, we demonstrate how we generate and update principles. Using the principles corresponding to X-ray cases as an example, Evo-PI utilizes five cases from the training set and then instructs the knowledgeable LLM to complete the generation of the principles.

When principles are updated, Evo-PI will provide the previously used principles and streamline and scale them.

Listing 1: Raw Principles Gereration Prompt

```
list some principles for these key medical reasoning tasks "X-Ray"

Here are the sample questions and answers:
    {
        "image": "Images/RadImageNet/bladder_pathology/abd132420.png",
        "problem": "What is the abnormality present in the image? A)
            Spinal cord injury B)Ovarian cyst C)Bladder pathology D)Liver
            cirrhosis",
        "solution": "<answer> C </answer>"
    },
    {
        "image": "Images/RadImageNet/normal/abd-normal028416.png",
        "problem": "Is there any deviation or anomaly observed in this
            image? A)No B)Yes.",
        "solution": "<answer> A </answer>"
    },
    {
        "image": "Images/RadImageNet/post_op/abd000807.png",
        "problem": "What type of abnormality is present in this image? A)
            Foreign body reaction B)Post-operative changes C)Infection D)
            Fracture site healing",
        "solution": "<answer> B </answer>"
    },
    {
        "image": "Images/RadImageNet/normal/abd-normal056990.png",
        "problem": "Is anything out of the ordinary evident in this image
            ? A)No B)Yes.",
        "solution": "<answer> A </answer>"
    },
    {
        "image": "Images/RadImageNet/interstitial_lung_disease/lung044245
            .png",
        "problem": "What type of abnormality is present in this image? A)
            Interstitial lung disease B)Pulmonary hypertension C)Asthma D
            )Pleural effusion",
        "solution": "<answer> A </answer>"
    },
and save it as a string list in Python.
```

Listing 2: Principles Updating Prompt

```
principles_prompt = [
    {"role": "system", "content": "You are helpful AI system in medical.\
        n\n"},
    {"role": "user", "content": f"Here are some key general principles
        for medical reasoning tasks involving {question_type}:\n\n"},
    {"role": "user", "content": f"{principles}\n\n"},
    {"role": "user", "content": "The principles should keep general and
        abstract, and as short as possible. If necessary, extend and
        merge the list of principles with new numbered principles;
        otherwise keep current principles unchanged. The listed
        principles should not exceed ten points The merged principles
        list:"}
]
```

### A.6 THE PROMPT OF GENERATE RAW PRINCIPLES

In this section, we will use the principles of X-ray as an example to demonstrate how these principles are specifically updated.

---

**Principles for X-ray in iteration 1**

1. Recognize common imaging signatures. X-rays typically show high-contrast grayscale images. Bone appears white, air (such as in the lungs) appears dark, and soft tissue is various shades of gray. Look for clear bony landmarks like ribs, spine, and clavicles.

2.Identify typical anatomical projections. Chest X-rays are often captured in posteroanterior (PA) or anteroposterior (AP) views. These images generally show the patient in an upright posture with visible lungs, diaphragm, and heart shadow.

3.Use process of elimination. Eliminate options that do not involve imaging (e.g., blood test, EKG). MRI images typically show high soft tissue contrast but have less visible bone detail. Ultrasound images have a grainy texture and do not show bones clearly. PET scans often include colorful overlays or are fused with CT, and do not resemble plain grayscale images.

4.Integrate clinical context. X-rays are commonly used as a first-line imaging tool for evaluating chest-related symptoms or conditions such as pneumonia, pneumothorax, or congestive heart failure.

5.Pay attention to image characteristics. X-rays are uniform grayscale images without the detailed cross-sectional anatomy seen in CT or MRI. They do not include high-resolution soft tissue differentiation or 3D rendering.

6.Rely on pattern recognition rather than memorization. Become familiar with the visual features that distinguish different imaging modalities. Recognize typical X-ray features through exposure to many examples.

---

**Principles for X-ray in iteration 2**

1. **Recognize Imaging Signatures:** Understand the high-contrast nature of X-rays, where bones appear white, air is dark, and soft tissues are gray.

2. **Understand Anatomical Projections:** Be familiar with common X-ray views like posteroanterior (PA) and anteroposterior (AP) to properly interpret anatomical structures.

3. **Differentiate Imaging Modalities:** Identify X-rays by their grayscale images and lack of high-resolution soft tissue contrast.

---

4. **Integrate Clinical Context:** Use X-rays in the context of clinical presentation to assess conditions like pneumonia, pneumothorax, and heart failure.

5. **Acknowledge Limitations:** Be aware that X-rays do not provide detailed cross-sectional or 3D images.

6. **Employ Pattern Recognition:** Develop proficiency through practice and focus on identifying typical X-ray patterns.

7. **Enhance Diagnostic Accuracy:** Combine X-ray findings with clinical information for a comprehensive evaluation.

---

**Principles for X-ray in iteration 3**

1. **Recognize Imaging Patterns:** Identify distinctive grayscale patterns for bones, air spaces, and soft tissues in X-rays.

2. **Understand Projections:** Be aware of common anatomical projections like PA and AP views for accurate image assessment.

3. **Differentiate Modalities:** Distinguish features of X-rays from those of MRI, CT, and ultrasound.

4. **Contextualize Clinically:** Integrate clinical data to enhance X-ray interpretation, aiding in diagnosis of conditions like pneumonia.

5. **Acknowledge Limitations:** Recognize the limitations of X-rays, such as limited soft tissue detail and no 3D rendering.

6. **Apply Pattern Recognition:** Develop the skill to efficiently recognize and interpret common X-ray findings.

7. **Ensure Systematic Approach:** Adopt a systematic approach to evaluating X-rays to ensure comprehensive analysis.

8. **Correlate with Findings:** Relate X-ray observations with clinical examination and laboratory results for a holistic view.

9. **Prioritize Safety:** Maintain awareness of radiation exposure and ensure justified use of X-ray imaging.

10. **Stay Informed:** Keep updated with evolving X-ray technology and guidelines for best practices.

## A.7 THE PROMPT OF GENERATE RESPONSES FORM MEDICAL MLLMS

For this section, we directly adopted the prompt provided on med-r1, thereby eliminating any sensitivity associated with the prompt.

---

**rollout prompt**

QUESTION_TEMPLATE = "Question First output the thinking process in `<think>B</think>` and final choice (A, B, C, D ...) in `<answer></answer>` tags."

## A.8 ADDITIONAL CASE STUDIES

For this section, we present case studies of different cases across various sub-datasets.

---

**Sample of CT case**

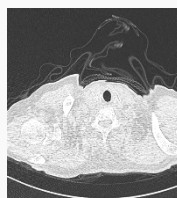

**Problem:** Is there any perceptible anomaly present in this picture?
  A)No    B)Yes
**Ground Truth:** A

**Med-r1's answer**:
<think> The image description A grayscale image of a cross-sectional view of a human head, showing the brain, skull, and surrounding tissues. The image appears to be a medical scan, possibly an MRI or CT scan, with various shades of gray representing different tissue densities. </think> <answer>B</answer>

**Evo-PI's answer (during iteration)**:
<think> The image provided is a CT scan of the neck, which is typically used to visualize the soft tissues and structures within the neck region. In a CT scan, different tissues appear in varying shades of gray, with bone appearing white, air appearing black, and soft tissues appearing in various shades of gray.
In this CT scan, the structures appear to be normal. The soft tissues around the neck, including the muscles and fat, appear to be in their expected positions and densities. There are no obvious signs of abnormal masses, calcifications, or other anomalies that would suggest a pathological condition.
Therefore, based on the visual inspection of the CT scan, there does not appear to be any perceptible anomaly present.
</think>
<answer>A</answer>

**Evo-PI's answer (evolved)**:
<think> 1. The image provided is a CT scan of the neck, which is typically used to visualize soft tissues and structures within the neck.

2. The scan shows a cross-sectional view of the neck, including the thyroid gland, which is a critical structure in the neck.

3. The thyroid gland appears to be in its normal position, with no visible signs of enlargement or abnormal shape.

4. There are no visible masses, calcifications, or other abnormalities that would suggest a pathological condition.

5. The surrounding structures, such as the trachea and carotid arteries, appear to be in their normal anatomical positions.

6. The image does not show any signs of inflammation, infection, or other conditions that would typically be indicated by abnormal findings on a CT scan.

Based on the above observations, there does not appear to be any perceptible anomaly present in the image.
</think>

---

```
<answer>A</answer>
```

**Sample of MI case**

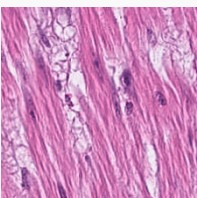

**Problem:** What is the name of the tissue shown in the picture?
   A)Nervous tissue    B)Cardiac muscle    C)Bone tissue    D)Smooth muscle
**Ground Truth:** D

**Huatuo-vision's answer:**
`<think>` TheThe image shows a histological section with elongated cells arranged in parallel bundles, which is characteristic of cardiac muscle tissue. The cells have striations and are organized in a way that suggests they are part of the heart's muscular structure.
`</think>` `<answer>B</answer>`

**Evo-PI's answer::**
`<think>` The image shows a microscopic view of tissue with distinct characteristics. The tissue appears to have elongated, striated fibers, which are typical of muscle tissue. The fibers are arranged in parallel bundles, and there are visible nuclei within the fibers, which is a common feature of muscle cells.
Let's analyze the options:
A) Nervous tissue: Nervous tissue typically consists of neurons and glial cells, which do not have the striated appearance seen in the image.
B) Cardiac muscle: Cardiac muscle is found in the heart and has a distinctive striated pattern, but it also has a different arrangement and structure compared to the tissue in the image.
C) Bone tissue: Bone tissue is composed of osteocytes embedded in a matrix of bone, which does not match the appearance of the tissue in the image.
D) Smooth muscle: Smooth muscle is found in the walls of blood vessels, the digestive tract, and other organs and has a different appearance compared to the tissue in the image.
Given the striated appearance and the parallel arrangement of fibers, the tissue in the image is most consistent with smooth muscle.
`</think>`
`<answer>A</answer>`

**Sample of MR case**

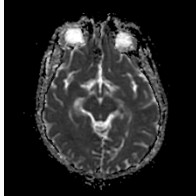

**Problem:** Is there anything unusual or irregular visible in this image?
   A)It's difficult to determine if there is an abnormality based on this image.
   B)There is a possibility of abnormality, but further tests are needed for confirmation.
   C)I'm not sure, but there might be a slight abnormality.
   D)No, It's normal.
**Ground Truth:** A

**Huatuo-vision's answer**:

`<think>` The image provided appears to be a brain MRI scan. In MRI scans, variations in signal intensity can indicate different tissue types or abnormalities. The bright areas could represent regions with increased water content, which may suggest edema or inflammation. However, without additional clinical context or comparative images, it is challenging to definitively state whether these findings are abnormal.

Given the options:
- A) It's difficult to determine if there is an abnormality based on this image.
- B) There is a possibility of abnormality, but further tests are needed for confirmation.
- C) I'm not sure, but there might be a slight abnormality.
- D) No, It's normal.

The most appropriate answer would be B) There is a possibility of abnormality, but further tests are needed for confirmation. This is because while the image shows some variations that could be indicative of an abnormality, a definitive diagnosis cannot be made solely from this single image without additional information and possibly further imaging or clinical correlation.
`</think>`
`<answer>B</answer>`

**Evo-PI's answer**:

`<think>` 1. Identify the Image Type and Purpose: The image is an MRI (Magnetic Resonance Imaging) scan of the brain, specifically an axial view (horizontal cross-section). This type of imaging is commonly used to visualize the brain's structures and detect abnormalities.

2. Analyze the Image Content:
- The image shows a clear, well-defined brain structure with no apparent distortions or irregularities.
- The white matter and gray matter are distinguishable, and there are no signs of abnormal hyperintensities (bright areas) or hypointensities (dark areas) that would suggest pathology.
- The ventricles and sulci (grooves) are visible and appear normal in size and shape.
- The skull and surrounding tissues are also clearly visible without any signs of abnormal thickening or thinning.

3. Consider the Context and Medical Knowledge:
- MRI scans are highly sensitive and specific for detecting abnormalities in the brain, including tumors, lesions, and structural anomalies.
- The absence of any abnormal signals or changes in the brain's structure in this image suggests that there are no immediate signs of pathology.
- The clarity and detail of the image indicate that it is a high-quality scan, which further supports the absence of any abnormalities.

4. Evaluate the Options:
- Option A: "It's difficult to determine if there is an abnormality based on this image." This option suggests uncertainty, which is not supported by the clear and detailed nature of the image.
- Option B: "There is a possibility of abnormality, but further tests are needed for confirmation." This option implies a need for additional testing, which is not indicated by the image alone.
- Option C: "I'm not sure, but there might be a slight abnormality." This option suggests uncertainty, which is not supported by the clear and detailed nature of the image.
- Option D: "No, It's normal." This option aligns with the clear and detailed nature of the image, indicating that there are no signs of abnormality.

5. Conclusion:

- The MRI scan of the brain shows no signs of abnormality, as evidenced by the clear and detailed structure of the brain, the absence of any hyperintensities or hypointensities, and the normal appearance of the ventricles and sulci.
- The high quality and clarity of the image further support the conclusion that there are no abnormalities present.

6. Final Analysis:
- The image does not show any signs of abnormality, and the clear and detailed structure of the brain suggests that the scan is normal.
- The options provided do not align with the clear and detailed nature of the image, which supports the conclusion that the scan is normal.

7. Final Decision:
- Based on the detailed analysis and the clear nature of the image, the correct answer is that there is no abnormality present, and the scan is normal.

Therefore, the final decision is that the image is normal, and no further tests are needed for confirmation.

```
</think> <answer>D</answer>
```

## A.9 DETAILED ABLATION EXPERIMENT

In this section, we present the results of ablation experiments using Med-R1 as the backbone MLLM. We observe that the average gain achieved by applying the principles reaches 13.48%. Building upon this foundation, incorporating iteration can further enhance performance by approximately 3.7%. On datasets like DER and US, which are relatively principle-dependent, the gains from starting with good initial principles are the greatest.

Table 3: Overall ablation study

| Dataset | CT | DER | FP | MI | MR | OCT | US | X-ray | Average |
|---|---|---|---|---|---|---|---|---|---|
| **Med-R1 [12] (Base [1])** | 0.7160 | 0.8338 | 0.9019 | 0.7447 | 0.5144 | 0.8946 | 0.7773 | 0.7854 | 0.7710 |
| **Evo-PI (Fix principles)** | 0.9278 | 0.9243 | 0.9140 | 0.8343 | 0.8174 | 0.9505 | 0.9955 | 0.8747 | 0.9048 |
| **Evo-PI (Iteration principles)** | 0.9676 | 0.9319 | 0.9195 | 0.8720 | 0.9462 | 0.9890 | 0.9982 | 0.9101 | 0.9412 |
| **Relative Gains from principles only (%)** | 21.18%↑ | 9.21%↑ | 1.21%↑ | 9.14%↑ | 30.30%↑ | 6.03%↑ | 21.82%↑ | 8.93%↑ | 13.48%↑ |
| **Relative Gains from iteration only (%)** | 3.98%↑ | 0.76%↑ | 0.55%↑ | 3.77%↑ | 12.88%↑ | 3.85%↑ | 0.27%↑ | 3.54%↑ | 3.70%↑ |
| **Relative Gains total (%)** | 25.16%↑ | 9.96%↑ | 1.76%↑ | 12.92%↑ | 43.18%↑ | 9.88%↑ | 22.09%↑ | 12.47%↑ | 17.18%↑ |

