# OpenReview forum: "Evo-PI: Scaling Medical Reasoning via Evolving Principle-Guided Reinforcement Learning"
_ICLR.cc/2026/Conference — ICLR 2026 Conference Withdrawn Submission_

### Official Review · Reviewer_tyP6 · 2025-10-27

**Soundness:** 2
**Presentation:** 2
**Contribution:** 2
**Rating:** 2
**Confidence:** 5

**Summary:**

1.  **Principle Bank Initialization:** A knowledgeable Large Language Model (LLM) generates an initial set of medical principles based on a few examples and category names (e.g., CT, X-ray).
2.  **Guided Reinforcement Learning:** The target MLLM is trained using RL (specifically, GRPO or GSPO are mentioned). A separate, frozen "judge" LLM evaluates the MLLM's reasoning trace against the current set of principles, generating a "principle reward" and a "thinking point reward". These are combined with standard RL rewards (accuracy, format) to update the MLLM.
3.  **Principle Evolution:** Based on the MLLM's performance and training dynamics (monitored via policy entropy), the knowledgeable LLM refines, expands, or prunes the principle set. This updated set is used in the next iteration.

**Strengths:**

The core concept of co-evolving principles and the reasoning model is highly novel and contrasts sharply with existing RLHF/RLAIF methods that use fixed reward models or heuristics. The idea of using abstract, evolving principles as reward signals is a significant departure.

**Weaknesses:**

1.  **Complexity and Reliance on External LLMs:** The framework involves a complex interplay between three models (knowledgeable LLM, judge LLM, MLLM backbone). The overall performance is heavily dependent on the quality and reliability of the knowledgeable LLM (for generating/evolving principles) and the judge LLM (for evaluating adherence). The paper uses powerful models (GPT-4O-mini, Qwen2.5-7B) but provides limited analysis on how sensitive the results are to the choice of these models or potential failure modes (e.g., the judge misinterpreting principle adherence, the knowledgeable LLM generating flawed principles).
2.  **Lack of Direct Principle Validation:** The "principles" are abstract and generated by an LLM. There is no external validation (e.g., by human medical experts) to confirm their clinical accuracy or relevance throughout the evolution process. The framework assumes the principles evolved by the LLM are inherently beneficial and correct.
3.  **Evaluation Limited to Accuracy:** While the motivation includes enhancing reasoning quality and clinical alignment, the primary evaluation metric is accuracy on VQA benchmarks. There is limited qualitative or quantitative analysis of the *reasoning traces* themselves to demonstrate improved clinical validity or reduced hallucination, beyond anecdotal case studies. How does Evo-PI compare on metrics specifically designed to evaluate reasoning faithfulness or factual consistency?
4.  **Potential for Principle Overfitting:** The principles evolve based on the MLLM's performance within the training loop. This creates a risk that the principles might overfit to the specific biases or failure modes of the MLLM or the training data distribution, rather than converging towards genuinely robust clinical heuristics.
5.  **Text-Only Judge:** The judge LLM evaluates principle adherence based only on the textual reasoning trace provided by the MLLM. It does not perform its own visual analysis. This means the rewards might reinforce textually plausible reasoning that is not actually grounded in the image evidence.
6.  **Lack of Benchmarks:** Why there is only OmniMedVQA used?

**Questions:**

1.  How robust is the framework to the choice of the knowledgeable LLM and the judge LLM? Have you experimented with different models (e.g., smaller open models, models with different medical domain expertise) for these roles, and how does it impact performance and the evolved principles?
2.  Could you provide more insight into the principle evolution process? How does the knowledgeable LLM decide *how* to refine principles based on "training dynamics" or "good/bad cases"? Is there a risk of generating incorrect or clinically misleading principles during this automated evolution?
3.  Beyond accuracy, have you evaluated the quality of the reasoning traces generated by the Evo-PI trained models? For instance, using metrics for factual consistency, hallucination rates, or even expert clinical review of sample reasoning paths?
4.  The judge LLM is text-only. How does the framework ensure that the reasoning steps rewarded for adhering to principles are actually grounded in the visual evidence, rather than just being textually plausible? Could a multimodal judge improve performance or reliability?
5.  The paper mentions mitigating reward hacking (Sections 1, 3.4, 3.5). Could you elaborate on the specific mechanisms within Evo-PI designed to prevent the MLLM from simply generating text that mentions principle keywords without demonstrating true understanding, especially given the judge is text-only?

---

> ### Author Response · Authors · 2025-11-20
>
> We sincerely thank the reviewer for this critical feedback.
>
> 1. Complexity and Reliance on External LLMs:
>
> We understand your concern regarding the complexity and reliance on external LLMs. Our approach aims to minimize external factors:
>
> Frozen Judge: We use a frozen judge of the same size from the same series as the backbone MLLMs. This minimizes the impact of other factors on our framework. The Frozen Judge approach does not involve external retrieval of text or knowledge; the inherent knowledge of existing medical MLLMs is already sufficient for its operation.
>
> Knowledgeable LLM's Role: Knowledgeable LLMs are used only to determine the quality of the initial principles. We will subsequently provide more options for knowledgeable LLMs and Judge models, along with related analyses, to demonstrate their impact.
>
> 2. Principle Validation:
>
> We agree with your assessment. One of the few assumptions in Evo-PI is that a knowledgeable LLM can initially provide a relatively reliable list of principles. Even if these abstract principles are not perfect, Evo-PI's iterative framework is designed to achieve significant performance improvements through refinement.
>
> 3. Evaluation Limited to Accuracy and reasoning traces:
>
> You've highlighted an important point about evaluation.
>
> Accuracy as Gold Standard: Accuracy remains the gold standard for determining whether a diagnosis is correct.
>
> Challenges in Reasoning Trace Evaluation: Accurately evaluating all reasoning traces still requires costly human experts. While automated methods exist, such as process-supervised reward models (PRMs) [1], current state-of-the-art medical PRMs achieve only about a 56% acceptance rate by human experts. This is a more challenging domain compared to fields like mathematics or coding, where verification is relatively easier.
>
> Future Plans: We have analyzed specific cases in our case study. In a future version, we plan to conduct a more comprehensive analysis of reasoning traces across all cases.
>
> [1] Process-Supervised Reward Models for Verifying Clinical Note Generation: A Scalable Approach Guided by Domain Expertise. Hanyin Wang, Chufan Gao, Qiping Xu, Bolun Liu, Guleid Hussein, Hariprasad Korsapati, Mohamad El Labban, Kingsley Iheasirim, Mohamed Hassan, Gokhan Anil, Brian Bartlett, Jimeng Sun
>
> 4. Potential for Principle Overfitting:
>
> We acknowledge the potential for overfitting within a single iteration.
>
> Entropy-Based Termination: We mitigate this by terminating training when entropy collapse occurs, indicating a lack of exploration.
>
> Iterative Refinement: The iterative refinement of the principles themselves helps prevent overfitting over the course of the entire training process by introducing new perspectives and adjustments.
>
> 5. Text-Only Judge:
>
> You raise a valid point about the judge's modality.
>
> Current MLLM Practices: Medical MLLMs typically employ processors to extract essential information from images and generate embeddings for inference when tackling medical VQA tasks [1]. The performance of current processors in backbone MLLMs is relatively reliable, effectively converting visual information into a textual or embedding format that a text-only judge can process.
>
> Future Considerations: We will also consider using MLLMs as the Judge. To avoid introducing additional external knowledge, we intend to use the backbone MLLMs themselves to perform the judging, leveraging their inherent multimodal capabilities without adding new variables.
>
> [1] Qwen2.5-VL Technical Report. Shuai Bai, Keqin Chen, Xuejing Liu, Jialin Wang, Wenbin Ge, Sibo Song, Kai Dang, Peng Wang, Shijie Wang, Jun Tang, Humen Zhong, Yuanzhi Zhu, Mingkun Yang, Zhaohai Li, Jianqiang Wan, Pengfei Wang, Wei Ding, Zheren Fu, Yiheng Xu, Jiabo Ye, Xi Zhang, Tianbao Xie, Zesen Cheng, Hang Zhang, Zhibo Yang, Haiyang Xu, Junyang Lin (additional authors not shown)
>
> 6. Lack of Benchmarks:
>
> OmniMedVQA encompasses diverse data from various examination types, providing a comprehensive evaluation under medical VQA. We plan to expand the datasets with additional relevant data in the future to further enhance its representativeness and challenge.
>
> 7. Robustness:
>
> Our design choices prioritize robustness and minimize confounding variables:
>
> Knowledge Distillation: The overall idea is to avoid distilling knowledge from powerful models to introduce more variables. The knowledgeable LLM only needs to provide abstract and reasonably reliable initial principles.
>
> Judge LLM Selection: For the judge LLM, we select an LLM of the same size from the same series as the backbone MLLMs. This prevents continuously distilling more powerful models during the judging operation, maintaining a consistent evaluation baseline.
>
> Future Robustness Investigation: We will consider using different Judge models and knowledgeable LLMs to further investigate the robustness of Evo-PI.

---

> ### Author Response · Authors · 2025-11-20
>
> 8. More insight into the principle evolution process:
>
> We appreciate the request for more insight into the principle evolution.
>
> Entropy-Based Refinement: In practice, we refine these abstract principles using knowledgeable LLMs by monitoring entropy trends [2]. Entropy reflects whether the model leans toward exploitation (using existing knowledge) or exploration (seeking new solutions). We refine the principles when entropy collapses, indicating the model is stuck in a local optimum.
>
> Theoretical vs. Practical Application: In theory, Evo-PI can update abstract principles using expert-selected good/bad cases as few-shot prompts. In practice, due to the lack of human expert evaluations in this domain, we have retained only the relevant implementation ways that do not require such expert input.
>
> [2] Know When to Explore: Difficulty-Aware Certainty as a Guide for LLM Reinforcement Learning.Ang Li, Zhihang Yuan, Yang Zhang, Shouda Liu, Yisen Wang
>
> 9. Mitigating reward hacking:
>
> We have two primary principles for mitigating reward hacking:
>
> Abstract Principles: To mitigate reward hacking, the principles are designed to be abstract, precluding the model from directly profiting by generating specific tokens and thus fostering a deeper, rather than superficial, understanding.
>
> Entropy-Based Truncation: The second involves determining whether to truncate the current iteration based on entropy conditions. This iterative property, combined with entropy monitoring, prevents reward hacking throughout the entire process by ensuring continuous learning and preventing the model from settling on suboptimal, reward-centric solutions.

---

### Official Review · Reviewer_SsZW · 2025-11-01

**Soundness:** 3
**Presentation:** 3
**Contribution:** 3
**Rating:** 6
**Confidence:** 3

**Summary:**

The paper proposes Evo-PI — a framework that teaches a medical VLM to reason through principle evolution. Instead of relying on a static reward model or handcrafted heuristics, Evo-PI keeps a bank of medical “principles” (abstract reasoning rules such as “recognize modality signatures,” “integrate clinical context,” etc.) and iteratively refines them through a knowledgeable LLM (GPT-4o-mini) after every RL phase.
Overall, I enjoy this idea having a evaluating knowledge database.

**Strengths:**

- The paper is very well written and clearly organized, making a complex reinforcement-learning framework easy to follow.

- I particularly appreciate the idea of iteratively expanding the knowledge base (principle bank) — this dynamic evolution of domain principles is both intuitive and elegant, and it mirrors how clinicians refine expertise over time.

- The method demonstrates strong and consistent performance improvements across multiple medical VQA benchmarks, showing that the evolving-principle strategy leads to tangible reasoning gains.

**Weaknesses:**

See questions.

**Questions:**

- My main concern is that the framework generates its knowledge base independently for each dataset and modality, resulting in separate sets of principles. While the model’s reported performance improves, it is unclear whether the underlying model itself becomes more capable, or if the gains simply reflect dataset-specific tailoring.

- From my understanding, the authors train and evolve principles per dataset modality. Would it not make more sense to train a single unified model using a combined dataset and a shared evolving principle bank, especially since the baseline comparisons (e.g., Med-R1, HuatuoGPT-Vision) are based on universal backbones?

- Code and model weights are not available. For a method this intricate—combining multiple LLMs, dynamic rewards, and iterative evolution—open-sourcing is essential for verification and reproducibility.

- I suspect the model could generate repetitive or redundant principles as a byproduct of its reward mechanism (“reward hacking”). While you may not need to empirically prove this, a verbal explanation or qualitative analysis of how Evo-PI prevents principle duplication or collapse would strengthen the paper.

---

> ### Author Response · Authors · 2025-11-20
>
> We sincerely thank the reviewer for this critical feedback.
>
> 1. For Validity:
> This is a very insightful question that touches upon the core of our framework's generalization capability. We thank the reviewer for raising this important point.
>
> Our current per-modality training approach was a deliberate design choice to isolate variables and clearly demonstrate Evo-PI's effectiveness across diverse and distinct medical tasks (e.g., analyzing cellular-level microscopy vs. anatomical CT scans). This allows for a clean and controlled evaluation.
>
> However, we agree with the reviewer that the ultimate goal is to enhance the model's intrinsic, generalizable reasoning capabilities. While our current experiments show that the backbone MLLM's parameters are updated and improved (as evidenced by accuracy gains), we acknowledge that training on a combined dataset with a shared, unified principle bank is a compelling next step.
> We believe such a unified approach could foster cross-modal knowledge transfer, where a principle learned from X-ray interpretation (e.g., 'identify high-contrast structures') could be adapted for CT scans. We will add a detailed discussion on this in the 'Future Work' section, highlighting it as a promising direction to build a more universally capable medical reasoning model.
>
> 2. For mixing datasets:
>
> We thank the reviewer for this excellent suggestion. Our decision to train and evolve principles on a per-modality basis was primarily methodological. As OmniMedVQA consists of distinct datasets for each modality, this approach allowed us to directly follow the evaluation protocol of our main baseline, Med-R1, ensuring a fair and controlled comparison.
>
> However, we strongly agree with the reviewer's intuition that training a single unified model on a combined dataset with a shared principle bank is a more powerful and scalable vision. This approach could unlock significant synergies and improve the model's core reasoning engine. We are very excited about this direction and will prominently feature it in our discussion of future work, outlining the potential benefits and challenges (such as managing potentially conflicting principles across modalities).
>
> 3. For reproducibility:
>
> We have uploaded the relevant code in the attachment, as mentioned in Reviewer xQku's Strengths 4.
>
> 4. For reward hacking:
>
> This is a very important concern. We address the risk of generating repetitive principles and reward hacking through a combination of mechanisms:
>
> a) Explicit Instruction in the Evolution Prompt: During the 'Principle Evolution' phase, the prompt given to the knowledgeable LLM explicitly instructs it to 'merge redundant rules, prune low-utility ones, and keep the principles abstract and concise.' This directly guides the evolution process towards quality rather than quantity.
>
> b) Dynamic Reward Signal: As the reviewer notes, the fact that the principles (and thus the reward function) evolve makes it a 'moving target' for the MLLM. It cannot simply memorize and repeat a few phrases to get a high reward, as the criteria for a good explanation are refined in the next iteration.
>
> c) Qualitative Example: For instance, in our X-ray experiments (shown in Appendix A.6), an initial, simple principle like 'Recognize imaging signatures' evolves into more specific and distinct principles in later iterations, such as 'Differentiate Modalities' and 'Ensure Systematic Approach.' This demonstrates a trend towards refinement and diversification rather than duplication.
>
> d) Entropy-based Stopping: Our use of policy entropy as a stopping criterion also helps. If the model's outputs become too repetitive (a sign of collapse/hacking), the entropy would drop sharply, triggering the termination of the training iteration.

---

### Official Review · Reviewer_xQku · 2025-11-07

**Soundness:** 2
**Presentation:** 2
**Contribution:** 2
**Rating:** 2
**Confidence:** 4

**Summary:**

The paper proposes Evo‑PI, a three‑stage pipeline for finetuning medical LLMs on medical VQA consisting of: (1) a Knowledgeable LLM distills “principles” from a few cases to initialize a principle bank; (2) a frozen judge LLM scores each rollout’s <think>…</think> reasoning trace against these principles to yield Principle Reward and Thinking‑Point Reward; (3) the principle set is iteratively refined and the process repeats. Training is done with RLVR (e.g., GRPO/GSPO). Core details: a frozen judge evaluates principle adherence and verifies step‑by‑step support for the final answer; rewards are formalized (Eqs. 3–4) and integrated into a PPO‑style objective (Eqs. 5–7); the rollout prompt enforces <think> and <answer> tags. The evaluation uses OmniMedVQA across eight modalities with Accuracy as the task metric; GPT‑4o‑mini is the Knowledgeable LLM and Qwen2.5‑7B‑Instruct is the judge.

**Strengths:**

1. Clear and sound algorithm formalization. The rewards are explicitly defined (Principle Reward / Thinking‑Point Reward) and combined with RLVR (GRPO/GSPO) in a standard clipped‑ratio objective with normalization.
2. Interpretable reasoning representation. The paper enforces <think>…</think> reasoning traces and uses them to compute verifiable‑format rewards and judge checks.
3. Consistent accuracy gains. On OmniMedVQA across eight modalities and two backbones (HuatuoGPT‑Vision, Med‑R1), the authors report sizable and consistent accuracy improvements (Table 2, §4.2).
4. Artifacts. Anonymized repo link and detailed prompts are provided in the appendix.

**Weaknesses:**

1. Fair Algorithmic vs. Poor Scientific Soundness

a) Algorithmic soundness (internal coherence). The optimization objective is well‑defined; reward clipping/normalization is specified; RLVR training with GRPO/GSPO is standard; iteration control via token‑level entropy is described. On these grounds, the method functions as designed.

b) Scientific soundness (external validity). The paper evaluates “reasoning quality” only via a frozen judge LLM (Qwen2.5‑7B‑Instruct) that compares traces against textual principles and “verifies, step by step” support for the answer; there is no clinician or human validation, no correlation of judge scores with expert ratings or clinical correctness, and no ground‑truth explanations. Thus, process‑quality claims remain self‑referential.

2. Evaluation scope is narrow for a medical‑reasoning claim
a) The paper reports accuracy as the sole task metric; it does not report process‑quality metrics against human judgement (faithfulness, plausibility, possible harm) or expert audits of traces/principles.

3. Unvalidated measurement instrument
a) The judge is frozen and unvalidated for clinical verification; there is no robustness or agreement study (e.g., alternative judges, judge–human correlations), which makes the central measurement instrument insufficiently trustworthy for claims about reasoning quality.

4. Clarity gaps:

a) Terminology confusion with Med PI vs Evo-PI. Also, there are many models in this process that don't have distinguishable names, this harms the clarity of Figure 2 discussed below.

b) Figure 2 (the three‑phase loop) is visually crowded, and component names overlap semantically; it’s hard to discern which boxes are fixed vs. learned. A simplified, story‑first schematic would help. (Figure appears with §3.2 discussion.)

c) Compute reporting lists hardware (4×H100 for training; 1×H100 for the judge) but omits GPU‑hours.

d) Table 1 lacks captions.

5. Mismatch between contribution and experiment design.

a) This work proposes a paradigmatic contribution, but the experiments don't measure the contribution of the new paradigm against the old (e.g., SFT). Instead, the result deltas only show that Evo-PI can finetune a previously finetuned model to excel on these benchmarks. However, by itself, this is hardly a surprising finding - if a model is finetuned on a benchmark, it is expected to improve its performance.

b) More appropriate experiments could include 1) starting from the baseline models' baseline models and comparing the finetuning impact under the baselines' paradigm vs Evo-PI, or 2) perhaps more easily, finetuning the baseline models under their respective previous paradigms can allow you to compare your existing results against their paradigms.

F. Fair Novelty.

a) The paper’s novelty framing ("Co-Evolution") is slightly exaggerated. Instead, the contribution is largely representational instead of paradigmatic as claimed—textual, evolving reward specification.

b) Relative to reward modeling (RLHF/RLAIF), Evo‑PI swaps a fixed neural reward regressor for an iteratively refined natural‑language principle set that a frozen judge LLM interprets. This improves transparency/interpretability but preserves the core policy–reward coupling; the novelty lies in representation and authoring of rewards, not in the RL optimizer itself. (This is an evaluative comparison; the paper does not present a RAG or RM baseline.)

**Questions:**

1. External validation: Can you run a small clinician audit (≈200 items, 2–3 raters) of reasoning traces (faithfulness, evidence‑grounding, plausibility, potential harm) and report inter‑rater agreement? (No human eval is currently reported.)

2. Judge calibration: Do judge scores correlate with human ratings and/or task Accuracy? Please report those correlations on a held‑out slice. (The judge is currently frozen and unvalidated.)

3. Robustness: How sensitive are results to the choice of judge (different families/sizes) and of the Knowledgeable LLM (GPT‑4o‑mini vs. open models)?

4. Causal use of steps: Can you add ablate‑a‑step and counterfactual occlusion tests to show the rewards drop when key evidence/steps are removed, demonstrating you’re incentivizing useful steps rather than verbosity?

5. Why principles vs. RAG? Please include a RAG baseline (textbook/diagnostic guideline retrieval) and optionally RAG+Evo‑PI to show whether evolving principles outperform or complement retrieval.

6. Compute: Please report GPU‑hours and wall‑clock per iteration (and #iterations).

---

> ### Author Response · Authors · 2025-11-20
>
> We sincerely thank the reviewer for this critical feedback.
>
> 1. Algorithmic soundness:
>
> There may be some misunderstanding. We want to demonstrate the powerful compatibility of Evo-PI as a framework.
>
> The Evo-PI framework we propose is compatible with various RLVR optimizers (such as GSPO), rather than being a single RL optimizer (such as GRPO). Evo-PI is compatible with various tricks, such as reward clipping. Evo-PI is also compatible with any form of reward model design, whether they are process-based or outcome-based rewards, such as format and ground rewards in GRPO.
>
> At its core, Evo-PI provides an extensible framework that enables scaling operations on supervised knowledge, which is manifested through these abstract principles.
>
> 2. Scientific soundness and evaluations:
>
> Evaluating reasoning traces is essentially a process reward model (PRM).
>
> It is precisely because human expert evaluation is prohibitively expensive that we introduce the concept of iterative abstraction principles as a substitute. During our evaluation, the ground-truth labels used to assess performance are unbiased, and this outcome reward model (ORM) not self-referential.
>
> Essentially, the Evo-PI judge model and these abstract principles, following their latest update, implicitly achieve a shift from unbiased ORM to biased PRM. This shifting can improve performance[1]. This is also the principle by which we further enhance backbone medical MLLMs without relying on human expert evaluations.
>
> For the reasons outlined above, we believe that human expert evaluations are valuable, feasible, and important, but not essential.
>
> Nevertheless, the full details of the human evaluation protocol and the robustness tests will be included in the subsequent version.
>
> [1] Amrith Setlur, Chirag Nagpal, Adam Fisch, Xinyang Geng, Jacob Eisenstein, Rishabh Agarwal, Alekh Agarwal, Jonathan Berant, Aviral Kumar: Rewarding Progress: Scaling Automated Process Verifiers for LLM Reasoning. ICLR 2025
>
> 3. Judge model and robustness:
>
> This is a crucial point. We acknowledge that the reliability of the judge LLM is central to our framework's validity. Our initial design choice was to use a same-series LLM to minimize external knowledge transfer, but we agree this choice needs validation. To test the robustness of our framework to the choice of the judge, we will conduct a new ablation study, which replace our original judge (Qwen2.5-7B-Instruct) with a model from a different family.
>
> 4. Clarity gaps:
>
> For a), b) and d): Thanks for pointing that out. We'll fix it in a future release.
> For c) Due to the use of different MLLM backbones and RL optimizers, GPU hours may vary. In these experiments, the worst-case scenario involved no more than 100 GPU hours.
>
> 5. Between contribution and experiment design:
>
> a) We have taken your concerns regarding the old paradigm into consideration.
> Therefore, we selected diverse backbones. For instance, HuatuoGPT-Vision employs SFT and alignment operations, while Med-R1 is an RL-optimized model (utilizing GPRO). The performance improvements on HuatuoGPT align with your perspective.
> Similarly, using the same RL optimizer on the already converged Med-R1 truly demonstrates the improvement of our framework. This precisely demonstrates the additional performance gains delivered by Evo-PI.
>
> b) Your suggestion is absolutely correct, and it is precisely what we have done. We loaded backbone medical MLLMs (such as Med-R1) into the Evo-PI framework and continued RL optimization to achieve the final results.
>
> In summary, Evo-PI is a framework rather than a model. It is compatible with multiple medical MLLMs and comprehensively enhances their performance. The backbone medical MLLMs have already provided a complete comparison between the old paradigms (e.g., SFT) and the new paradigms (e.g., RL), and this is not the focus of our discussion.

---

> ### Author Response · Authors · 2025-11-20
>
> 6. Fair Novelty:
>
> a) The co-evolution paradigm mentioned in Evo-PI fundamentally demonstrates that scaling up external abstract knowledge remains effective and powerful. Knowledge is stored in the form of principles, and scaling operations are implemented through iterative processes.
>
> b) Using the same size of frozen judge from the same series minimizes the impact of other factors on the framework.
> The Frozen Judge approach does not involve query-related external retrieval of text and knowledge. Abstract principles may resemble the high-level task descriptions typically found in system prompts. However, Evo-PI does not explicitly convey them through prompts; instead, they are implicitly embedded within the Judge model. Hence,comparisons with other RAG-based methods are unnecessary. The performance of existing medical MLLMs as baselines demonstrates that their inherent knowledge is already sufficient. For the same reason, after introducing other RMs, backbones implicitly distill RMs' knowledge. Our design does not benefit from such implicit distillation, meaning our performance gains primarily rely on the design of the Evo-PI framework.
>
> 7. External human validation and calibration:
> Detailed discussion is shown in the "Scientific soundness and evaluations". We believe the accuracy assessment is essential, as it reflects the proportion of misdiagnoses, and human evaluation and annotation remain quite costly. We will open-source our model weights and results for community evaluation.
>
> 8. Robustness of Judge model and Knowledgeable LLM:
> The insights behind the design of the judge model have been presented above. Regarding Knowledgeable LLM, the principles it generates undergo iterative updates, so they may not be particularly sensitive. This is quite interesting, and we will provide more analysis in future versions.
>
> 9. Causal use of steps:
> In Evo-PI experiments, Med-PI serves as the backbone MLLM, which is then further optimized using GRPO. This demonstrates the effectiveness of our Evo-PI approach. Ablation experiments for the steps will be provided in future versions.

---

### Official Review · Reviewer_YAUF · 2025-11-10

**Soundness:** 3
**Presentation:** 2
**Contribution:** 3
**Rating:** 4
**Confidence:** 4

**Summary:**

The paper proposes an Evolving Principle–guided reinforcement learning framework (Evo-PI) for medical visual question answering. Evo-PI externalizes domain knowledge into an editable principle bank, uses a frozen judge LLM to convert principle adherence and “thinking-point” checks into verifiable rewards, and iteratively scales/refines the principles across training rounds. Experiments are conducted on the OmniMedVQA benchmark across eight imaging modalities.

**Strengths:**

1. The target problem is valuable.
2. The experiments on a benchmark show that the proposal achieves consistent improvements on eight imaging modalities.

**Weaknesses:**

**Major:**

1. The claim of evaluation on “eight medical VQA benchmarks” misleads readers. In fact, the paper only performs experiments on a single benchmark dataset OmniMedVQA (Section 4.1).

2. The paper should extend the experiments to more widely used benchmark datasets, such as: (1) general medical VQA datasets VQA-RAD, SLAKE, PathVQA, and PMC-VQA; (2) medical reasoning benchmarks MMMU (H&M) (Yue et al., 2024) and MedXpertQA (Zuo et al., 2025).

3. The evaluation metric is limited. The paper claims that the proposed method can generate trustworthy, scalable, generalizable, robust, clinically aligned, and expert-like reasoning. However, the paper only reports accuracy for evaluation. Moreover, the usefulness of “principle-guided” explanations in practice should also be assessed. Robustness and OOD stress-testing should be provided as well [Beyond Benchmarks: Dynamic, Automatic And Systematic Red-Teaming Agents For Trustworthy Medical Language Models, arXiv 2025].

4. More analysis should be provided. The paper reports average gains but does not vary: (i) the number, length, or abstraction level of principles; (ii) iteration count; (iii) judge model size/type; (iv) knowledgeable LLM choice; or (v) masking of the <think> channel. These factors are crucial to cost-effectiveness.

**Minor:**

5. Limited backbone diversity. Although the paper claims the approach is general, both backbones are Qwen-family VL models (7B and 2B variants), and the judge is Qwen2.5-7B-Instruct.

6. Statistical significance tests should be provided, considering the limited evaluation data.

**Questions:**

Please see Weaknesses

---

> ### Author Response · Authors · 2025-11-20
>
> We sincerely thank the reviewer for this critical feedback.
> 1. Clarification on OmniMedVQA:
> We acknowledge that our phrasing 'eight medical VQA benchmarks' was imprecise and could be misleading. We will revise the manuscript to clarify that we evaluated our method on a single comprehensive benchmark, OmniMedVQA, which spans eight distinct medical imaging modalities. Our intention was to demonstrate the versatility of our approach across different data types within a unified framework.
> 2. Experiments on Additional Datasets:
> To address the crucial point about generalization to more widely-used benchmarks, we will consider adding the reasoning datasets you provided in future versions.
> 3. Evaluation metrics:
> We agree with the reviewer that accuracy alone is insufficient to validate our claims about generating trustworthy and expert-like reasoning. This is a critical point, and we will take steps to address it. The full details of the human evaluation protocol and robustness tests will be added to the following version.
> 4. Extra factor analysis:
> Thank you for suggesting these important ablation studies. We agree that a deeper analysis is crucial for understanding the framework's dynamics and cost-effectiveness. i, iv, v) Analysis of Principles and LLM Choices: We acknowledge this limitation. While a full quantitative analysis is complex, we have added a qualitative analysis in the appendix. We show how principles for the X-ray modality evolve over iterations, becoming more specific and clinically nuanced. We also discuss that while we used GPT-4o-mini, the framework is flexible, and the choice of knowledgeable LLM primarily impacts the quality of the initial principle bank. ii) Impact of Iteration Count: Our ablation study in Appendix A.9 (Table 3) already provides a partial answer. The 'Relative Gains from iteration only' row shows that moving from fixed principles to iterative principles (which implies multiple iterations) provides an average performance boost of 3.7%. We will add a plot showing performance vs. iteration count (1, 2, 3) to make this clearer. iii) When designing the judge model, I prefer to avoid introducing additional information to prevent the Med-PI from relying on external knowledge. Therefore, we employ a pure text LLM (e.g., Qwen 2.5-7B in our paper) from the same series and with the same number of parameters, trained on the same knowledge source. We will add this to our ablation studies.
> 5. Backbone diversity:
> We appreciate the reviewer's concern. Our choice of the Qwen family was motivated by its state-of-the-art performance and our desire to control for architectural variables when comparing our method against the baselines. We agree that demonstrating performance on different model families (e.g., LLaVA-based) would further strengthen our generality claim, and we will explicitly state this as a limitation and a direction for future work in the paper.
> 6. Statistical Significance Tests:
> Our reported results in Table 2 represent averaged performance across all runs , comparing the performance of the backbones before and after applying Evo-PI. In the final manuscript, we will also report the corresponding variance to illustrate performance fluctuations.

---

### Note · Authors · 2026-01-05

I have read and agree with the venue's withdrawal policy on behalf of myself and my co-authors.